# The Effect of Downsizing Packages of Energy-Dense, Nutrient-Poor Snacks and Drinks on Consumption, Intentions, and Perceptions—A Scoping Review

**DOI:** 10.3390/nu14010009

**Published:** 2021-12-21

**Authors:** Qingzhou Liu, Lok Yin Tam, Anna Rangan

**Affiliations:** Nutrition and Dietetics Group, School of Life and Environmental Sciences at the Charles Perkins Centre, The University of Sydney, Sydney, NSW 2006, Australia; qingzhou.liu@sydney.edu.au (Q.L.); ltam8308@uni.sydney.edu.au (L.Y.T.)

**Keywords:** downsizing, eating behaviour, energy-dense nutrient-poor foods, snacks, package size effect, portion size

## Abstract

The single-serve packaging of discretionary foods is becoming increasingly popular, but evidence is limited on whether smaller package sizes can reduce food intake. The aim of this scoping review is to assess the effect of reducing the package size of energy-dense, nutrient-poor (EDNP) snacks and drinks on consumption, intentions, and perception, and to examine the effects of potential moderators or mediators. The search was conducted in six selected databases and grey literature sources, following the Preferred Reporting Items for Systematic Reviews and Meta-Analyses for the scoping review process (PRISMA-ScR) guidelines. After screening 5562 articles, 30 articles comprising 47 intervention studies were included. Twelve of 15 studies found a significant effect in lowering the actual or intended consumption when a single smaller package was offered compared with a single larger package. When the total serving size was held constant between varying package conditions, such as a multipack, single package, or unpackaged, the results on the actual and intended consumption were inconsistent and varied according to the presence of moderators. Overall, these findings suggest that an overall reduction in the size of a single package is a more promising strategy than providing multipacks to reduce consumption. Changes to the current food environment to promote single smaller packages of EDNP snacks and drinks are necessary to support the better selection of appropriate portion sizes and reduce consumption.

## 1. Introduction

Excessive energy intake is one of the key drivers of the worldwide obesity epidemic. Energy-dense, nutrient-poor (EDNP) foods and drinks that are high in saturated fat, added sugars, salt, and/or alcohol are excessively consumed and contribute to increased energy intake, weight gain, and overweight and obesity [1,2,3,4]. In Australia, EDNP foods account for over one-third of the total energy intake among all age groups, more than double the recommended amount [2,5]. Similar patterns have been observed in other western countries such as the United Kingdom (UK) and the United States of America (USA) [6,7].

The typical portion sizes of many EDNP foods and drinks have increased over time. For example, in Australia, ice-cream portion sizes have increased 39% between national surveys conducted in 1995 and 2011–2012 [3]. Data from the USA, Europe, and New Zealand reflect similar trends and demonstrate that the serving (the amount of food or drink provided or served for consumption) and portion (the amount of food or drink selected or consumed by an individual at any particular eating occasion) sizes of numerous EDNP foods have increased significantly both in-home and out-of-home [8,9,10,11]. Food companies profit from selling large servings, as these are seen as better value for money by consumers, with often only a small price differential between small and large servings [12]. The widely promoted and marketed EDNP foods, and the ubiquity of large serving sizes of these foods and drinks at a relatively low cost creates a food environment that encourages overconsumption [12].

There is considerable evidence that the serving size or package size acts as an environmental cue to influence the amount of food consumed [13,14,15,16]. People tend to eat more when presented with a larger serving size, coined the ‘portion size effect’, and this has been demonstrated in both experimental and real-life settings [13,14,16]. Single-serve packaging of EDNP snacks and drinks is becoming increasingly popular [17,18], and similar to the portion size effect, the ‘package size effect’ refers to larger package sizes leading to increased consumption [14]. The package size can act as a prompt that alters consumers’ perception of a normal size [14,19]. Constant exposure to large package sizes can reset norms of what an appropriate size should be [20]. The presence of a large number of moderators and mediators also affects eating behaviours when people are exposed to various-sized packaging [13,14,21]. For example, when presented with a larger serving size, adults tend to overconsume to a greater extent than children [14]. The type of food is also a potential moderator of the package size effect, with people tending to consume greater amounts of highly palatable EDNP foods than healthier core foods when exposed to a larger serving size [14,19]. To add to this complexity, numerous moderators and mediators appear to interact with each other, contributing to individual differences in susceptibility to large servings and packages [21,22].

As an increase in portion sizes has been linked to excessive weight gain due to little energy compensation at subsequent meals [3,13,23], recent reviews recommend targeting large portion and package sizes as a potential strategy to help mitigate the rising trend in overweight and obesity [19,22]. A review by Hetherington and colleagues indicated that downsizing is a potential solution to prevent the excessive intake of energy-dense food, as well as to adjust expectations of ‘appropriate’ amounts of food among children and adolescents [22]. Similarly, Almiron-roig and colleagues suggested that more effective population-wide policies are needed to restrict accessibility to large serving and package sizes [19]. However, evidence on the effect, acceptability, and the feasibility of downsizing packages of EDNP snacks and drinks is currently limited.

A comprehensive overview on the role of package sizing in determining consumption and how consumers respond to package size reduction is essential to find practical solutions for better portion control of EDNP snacks and drinks at the population level. The objective of this scoping review is to systematically map the existing research studies to (1) assess the effect of reducing the package size of EDNP snacks and drinks (alcoholic and non-alcoholic) on consumption, intention to consume and purchase, and perceptions; (2) assess the effects of potential moderators or mediators. These findings might contribute to the evidence base for developing recommendations and policies to optimise serving sizes for EDNP snacks and drinks.

## 2. Materials and Methods

This review was conducted in accordance with the Preferred Reporting Items for Systematic Reviews and Meta-Analyses for the scoping review process (PRISMA-ScR) and the Joanna Brigg’s Institute (JBI) scoping review methodology [24,25]. The review protocol was registered with the Open Science Framework (registration DOI: 10.17605/OSF.IO/SQNVK).

### 2.1. Eligibility Criteria

The eligibility criteria of this review are summarised in Table 1 using the Participant, Concept, Context (PCC) framework.

The following terminology was used in this review [26,27]: Portion size refers to the amount of food or drink selected or consumed by an individual at any particular eating occasion [27]. (Total) serving size refers to the (total) amount (volume or weight) of food or drink that is served for consumption [27]. Package size refers to the volume of packaging (for example, can, bottle, pouch, box), in which a specific amount of food or drink is present [14,26]. Single pack or package refers to a single packaging unit provided (or sold) to the consumer that contains food or drink [26]. Multipacks refer to a group of multiple small single packs provided (or sold) to the consumer [26]. Energy-dense and nutrient-poor snacks and drinks include potato crisps, sweet biscuits, cake, ice cream and other ice confections, sugar-sweetened drinks, and other snack foods and drinks that are high in saturated fat, added sugar, salt, and/or alcohol [1,28].

### 2.2. Information Sources and Search

Search strategies for selected databases and grey literature sources were developed in collaboration with an experienced liaison librarian. To identify and map all potentially relevant studies, the search strategy was performed in six selected databases (from inception to February 2021), including MEDLINE, CINAHL, Embase, PsycInfo, ABI/INFORM, and Business Source Ultimate. Filters were only applied to the ABI/INFORM databases to exclude wire feeds, webpages, newspapers, and company reports, as they were deemed irrelevant to this review. To identify grey literature, a full search strategy was performed in ProQuest Dissertation & Thesis, and a forward citation search was undertaken using the most comprehensive reviews identified from the preliminary search [14,29]. Upon completion of full-text screening, backward and forward hand searching was performed by scanning through the reference lists and ‘cited by’ lists for all the eligible studies. The full search strategy for the MEDLINE database is attached (Appendix A).

### 2.3. Study Selection

The study selection process was performed by two reviewers (Q.L. and L.Y.T.) independently using the reference management software, Endnote X9 [30]. Study retrieval involved three steps: (1.) Identification of studies in all databases and grey literature and removal of duplicates; (2.) title and abstract screening; (3.) full-text screening. A calibration exercise of the first 50 citations was performed prior to the formal title and abstract screening process to ensure consistency of study selection between reviewers. Any discrepancies were discussed between the two reviewers first, and further discussion with the third author was undertaken if consensus was not reached.

### 2.4. Data Extraction and Results Presentation

Data from each included study was charted by two reviewers (Q.L. and L.Y.T.) independently into a pre-designed data-charting template using Microsoft Excel software. Summarised data included study sample (number of participants, age, gender), intervention and comparison groups, potential moderators or mediators, outcome measure, and findings. Articles using the same intervention, but different outcome measures were counted as separate studies in this review [31,32]. Studies were grouped into three broad categories according to package size interventions: (1.) smaller versus larger single pack with different total serving size; (2.) smaller multipacks versus larger pack(s) containing same total serving size; (3.) presence versus absence of package/wrapping, containing same total serving size. A moderator is defined as variable that influences the direction and/or strengths of the relation between the independent variable and the dependent variable; a mediator is defined as a variable that links the independent variable and the dependent variable, and the existence of mediator explains the relationship between the other two variables [33,34]. More detailed information on study design, setting, and potential moderators and mediators are provided in Appendix A.

### 2.5. Critical Appraisal

The quality of each eligible study was assessed using The Joanna Briggs Institute’s Critical Appraisal Checklists for the appropriate study type [35]. The risk of bias of individual intervention studies was classified as either high (if three or more criteria were assessed as No or Unclear) or low risk of bias (if less than three criteria were assessed as No or Unclear) [36].

## 3. Results

This section may be divided by subheadings. It should provide a concise and precise description of the experimental results, their interpretation, as well as the experimental conclusions that can be drawn.

### 3.1. Selection of Sources of Evidence

The final search yielded a total of 5562 research articles (5072 from the selected databases and 490 from grey literature sources). After removing duplicates (*n* = 2271), 3291 articles remained for the title and abstract screening. After excluding 3216 articles, 75 full-text articles were assessed against the eligibility criteria. Of these, 26 articles were identified as meeting the eligibility criteria, and hand searching of these 26 articles identified four more eligible articles, resulting in a total of 30 articles comprising of 47 intervention studies (Figure 1). The PRISMA-ScR Checklist is attached (Appendix A).

### 3.2. Study Characteristics and Critical Appraisal

Within the 30 eligible articles, 47 separate intervention studies were reported (may have different food, package size, or outcome). All intervention studies were conducted in high-income western countries, with the majority conducted in the USA (*n* = 22), followed by the Netherlands (*n* = 7), UK (*n* = 6), Belgium (*n* = 5), Canada (*n* = 4), and Australia (*n* = 3). Figure 2 illustrates the publication years and countries of the included articles. Most of the studies were randomised–controlled trials (RCT) (*n* = 26) or quasi-experimental studies (*n* = 20), with the exception of one qualitative study. The majority of studies used face-to-face or computer-based laboratory settings (*n* = 35), eight studies used free-living settings (where participants were able to take provided foods or drinks home for consumption), and four studies were conducted in naturalistic settings (such as classrooms or movie theatres). The sample size of most studies varied between 50 and 300 participants. Study participants were predominantly university students or adults in their 20–30s (*n* = 39). Two studies were conducted in young children aged between 3 and 7 years. Four studies recruited females only, the remaining studies recruited both males and females. The most commonly examined EDNP snacks and drinks were confectionery (*n* = 20), biscuits and sweet pastries (*n* = 11), sugar-sweetened drinks (*n* = 5), alcohol (*n* = 3), popcorn (*n* = 3), mixed snack boxes (*n* = 3), and savoury snacks (*n* = 2).

Based on The Joanna Briggs Institute’s Critical Appraisal checklists, 21 studies were rated as a low risk of bias (seven studies in Table 2, ten studies in Table 3, and four studies in Table 4) and 26 were rated to be at a high risk of bias. Most of the reviewed studies met the criteria for adequate and reliable outcome measurements, sufficient follow up, and appropriate statistical analysis. The majority of RCTs did not provide sufficient information relating to the randomisation method and blinding of intervention group allocation. For quasi-experimental studies, most studies did not have a control group and failed to compare participant characteristics between comparison groups. A summary of the risk of bias assessment is attached (Appendix A).

### 3.3. Package Size Effects, Moderators, and Mediators

The package size intervention studies are summarised according to study design in Table 2, Table 3 and Table 4.

**Table 2 nutrients-14-00009-t002:** Summary of intervention studies comparing a smaller versus larger single pack containing different total serving size of energy-dense, nutrient-poor snack and drink.

First Author, Year of Publication, Country, Risk of Bias	Study Sample	Setting	Package Size Comparison Groups	Potential Moderators or Mediators	Outcome Measures, (Measures Used)	Findings
Aerts, 2017 Study 1 [37] Belgium Low	96 (46 girls) Mean age 6.4 ± 0.7 years	Naturalistic (school classroom)	Popcorn (sugared or salted) 60 g bucket 30 g bucket	Food preference (sugared/salted) Age Gender	Consumption (direct weighing)	Children consumed significantly more (24 g/89%) from the larger pack than smaller for both sugared and salted popcorn. The tendency to overconsume from the larger pack was higher when served sugared popcorn (preferred) than salted popcorn. Age and gender were not moderators.
Aerts, 2017 Study 2 [37] Belgium Low	55 (26 girls) Mean age 4.7 ± 0.9 years	Naturalistic (school classroom)	Cookies 48 g box 30 g box	Age Gender	Consumption (direct weighing)	Children consumed significantly more (7 g/30%) cookies from the larger pack than smaller pack. Age and gender were not moderators.
John, 2017 Study 2 [38] The USA Low	470 (211 females) Mean age 33 years	Laboratory computer-based	Sugary drinks (iced tea or lemonade) 680 mL cup 454 mL cup	None	Consumption (direct weighing) Likelihood of purchase (computer task)	Participants who purchased a smaller-sized drink consumed significantly less than those who purchased a larger-sized drink. The likelihood of purchase between drink sizes did not differ.
John, 2017 Study 3a The USA [38] High	557 (261 females) Mean age 32 years	Laboratory computer-based	Sugary drinks (iced tea or lemonade) 567 mL cup 454 mL cup	None	Consumption (direct weighing) Likelihood of purchase (computer task)	Participants who purchased a smaller-sized drink consumed significantly less than those who purchased a larger-sized drink. The likelihood of purchase between drink sizes did not differ.
Marchiori, 2012 [39] Belgium Low	88 students (62 females) Mean age 20.1 ± 2.1 years	Laboratory face-to-face	M&M’s 600 g box 200 g box	Age Food preference Weight	Consumption (direct weighing)	Participants in smaller (200 g) box condition consumed 30 g/150 kcal (50%) less than those in larger (600 g) box condition. Age, food preference, and weight were not moderators.
Rolls, 2004 [40] The USA High	60 (34 female) Mean age 22.9 years	Laboratory face-to-face	Potato chips 170 g, 128 g, 85 g, 42 g, 28 g bag	Gender Age Dietary restraint status Weight	Consumption (direct weighing)	Participants of both genders consumed significantly less (females 184 kcal less, males 311 kcal less, from the largest to smallest packages) when the package size was incrementally reduced from 170 g to 28 g. This effect was more prominent for males than females. Age, dietary restraint, and weight were not moderators.
Versluis, 2016 Study 2 [41] The Netherlands Low	224 university students (92 females) Mean age 21 ± 1.6 years	Laboratory face-to-face	M&M’s 400 g bag 200 g bag	Diet prime (commercials) ^1^ Dietary restraint status Food preference Gender Weight	Consumption (direct weighing)	No significant effect of package size on consumption was found. When exposing to a diet prime prior to eating, restrained eaters consumed significantly less from larger pack, but not from smaller pack. Exposing to a diet prime prior to eating did not influence consumption in unrestrained eaters. Dietary restraint, food preference, gender, and weight were not moderators.
Wansink, 2001 [42] The USA High	151 moviegoers (66 females) Age range 11–89 years	Naturalistic (movie theatre)	Popcorn 240 g bucket 120 g bucket	Food preference (perceived taste)	Consumption (direct weighing) Perception (healthiness) (questionnaire)	Participants consumed significantly less (33 g/35%) from the smaller pack than larger pack. This effect was more prominent in participants who rated the taste as favourable than those who rated the taste as unfavourable. Participants tended to pay more attention to monitor their intake when eating from the smaller pack, and they perceived popcorn in the smaller pack to be healthier than from the larger pack.
Wansink, 2005 [43] The USA High	158 moviegoers (67 females) Mean age 28.7 years	Naturalistic (movie theatre)	Popcorn 240 g bucket 120 g bucket	Food preference (fresh/stale)	Consumption (direct weighing)	Participants consumed significantly less (20 g/28%) from the smaller pack than larger pack. This effect was more prominent for the fresh popcorn (preferred) than for the stale popcorn.
Clarke, 2020 The UK [44] Low	140 (96 females) Mean age 41 years	Laboratory face-to-face	Wine 750 mL bottle 500 mL bottle	Gender	Intention to consume (self-selection using real food)	No effect of wine bottle size on intention to consume was found. Gender was not a moderator.
Versluis, 2015 Study 1 [45] The Netherlands High	317 (159 females) Mean age 44 ± 12 years	Laboratory computer-based	Milk chocolate 180 g bar 75 g bar	Gender Serving size recommendation labelling (pictorial) ^2^	Intention to consume (computer task)	Participants intended to consume significantly less (11 g/56 kcal (22%)) from the smaller pack than larger pack. This effect was only significant among males. Serving size recommendation labelling was not a moderator.
Versluis, 2015 Study 2 [45] The Netherlands High	324 (154 females) Mean age 38 ± 11 years	Laboratory computer-based	Milk chocolate: 180 g vs. 75 g M&M’s: 400 g vs. 165 g Crackers: 120 g vs. 60 g	Gender Serving size recommendation labelling (pictorial)	Intention to consume (computer task)	Participants intended to consume significantly less (22 g/27%) from the smaller pack than larger pack. This effect was significant for both genders, but it was more prominent for males than for females. The pictorial serving size recommendation labelling resulted in lower intention to overconsume when package size was large but not when small.
Versluis, 2016 Study 1 [41] The Netherlands Low	477 (244 females) Mean age 40 ± 11 years	Laboratory computer-based	Milk chocolate: 180 g vs. 75 g M&M’s: 400 g vs. 165 g Potato chips: 300 g vs. 120 g	Diet prime (health magazines) ^3^	Intention to consume (computer task)	Participants who were exposed to non-diet prime (travel magazine, as the control group) prior to eating had significantly lower intention to consume from the smaller pack than larger pack. Exposing to diet prime prior to eating diminished this effect, no difference in intention to consume between the smaller and larger pack was found.
Wansink, 1996 Study 4 [46] The USA High	184 females 39 participants completed the follow-up questionnaire	Laboratory computer-based	M&M’s Large bag: 342 chocolates Medium bag: 228 chocolates Small bag: 114 chocolates	None	Intention to consume (self-selection using real foods) Perception of snack unit prices (face to face survey)	Participants intended to consume significantly more (40 g/63%) from the medium pack than small pack. Participants intended to consume significantly more (59 g/94%) from the large pack than small pack. No significant difference in intention to consume was found between the medium- and large-sized packs. Participants perceived the unit price to be higher when package sizes became smaller.
Huyghe, 2013 [47] Belgium High	235 (157 females) Mean age 32.4 ± 13.8 years	Laboratory computer-based	Cookies, muffin, chocolates, chocolate bar 40 g, 80 g, 120 g, 160 g, 200 g, 240 g, 280 g, 320 g	Gender	Intention to purchase (computer task)	No effect of snack package size on intention to purchase was found. Gender was not a moderator.

^1^ Diet prime: diet-related commercials with messages focused on resisting temptation of foods (for example, dieting, setting and reaching goals, weight loss plan); non-diet prime (control group): non-diet-related commercials, no message related to dieting, food, or exercise. ^2^ Pictorial serving size recommendation labelling: using picture of snack food in nutrition labelling (e.g., a picture of four pieces of chocolates as the recommended serving size), which is different from non-pictorial labelling that uses text. ^3^ Diet prime: health magazine with messages related to weight loss, diets, and fitness; non-diet prime (control group): travel magazine, no message related to dieting, food, or exercise.

**Table 3 nutrients-14-00009-t003:** Summary of intervention studies comparing smaller multipacks versus larger package(s) containing same total serving size of energy-dense, nutrient-poor snack and drink.

First Author, Year of Publication, Country, Risk of Bias	Study Sample	Setting	Package Size Comparison Groups	Potential Moderators or Mediators	Outcome Measures (Measures Used)	Findings
Argo, 2012 Study 2 [48] Canada Low	207 undergraduate students (123 females)	Laboratory face-to-face	Candy-coated chocolates Two larger packs Eight smaller packs	Package design (transparent/opaque) Appearance self-esteem (ASE) ^1^ Gender	Consumption (direct weighing)	Participants consumed significantly more from the smaller multipacks than larger packs, which was fully contributed by those with low ASE. No effect was found among those with high ASE. When packaging was transparent (vs. opaque), participants consumed significantly more (42 g/100%) from the smaller multipacks than larger packs. Gender was not a moderator.
Bui, 2017 Study 3 [49] The USA Low	67 undergraduate students (35 females) Mean age 27 years	Laboratory face-to-face	Bite-sized chocolate chip cookies One larger pack (16 pieces per pack) Four smaller packs (4 pieces per pack)	Gender	Consumption (direct weighing)	No significant effect of package size on consumption was found. Gender was not a moderator.
Codling, 2020 [50] The UK Low	166 households Mean age 31 years	Free living	Wine 750 mL bottle 500 mL bottle	The order of receiving each package condition (crossover)	Consumption (recording empty bottles)	Participants (households) consumed significantly less wine in 14 days (173 mL/4%) and had a lower rate (6%) of consumption from the 500 mL bottles than 750 mL bottles. The order of receiving each package size condition was not a moderator.
Do Vale 2008 Study 2 [51] The Netherlands High	140 undergraduate students (59 females)	Laboratory face-to-face	Potato chips Two 200 g packs Nine 45 g packs	Self-regulatory concern ^2^	Consumption (direct weighing)	No significant effect of package size on consumption was found. The activation of self-regulatory concern led to lower intake from larger packs (but not from smaller multipacks).
Haire, 2014 [52] The USA High	64 university students (30 females) Mean age 23.7 years	Free living	Mini-pretzel Two 283 g packs Twenty-two 26 g packs	Weight (22.2 kg/m^2^ in normal weight group; 29.8 kg/m^2^ in overweight group) Dietary restraint status	Consumption (direct weighing)	Overweight or obese participants consumed significantly less (97 g/361 kcal (48%)) from the smaller multipacks than larger pack. No significant effect of package size was found among normal weight participants. Dietary restraint was not a moderator.
Holden, 2015 Study 1 [53] Australia Low	108 university students (58 females)	Laboratory face-to-face	M&M’s One 200 g pack Four 50 g packs	Manipulated diet consciousness ^3^ Measured diet consciousness ^4^	Consumption (direct weighing)	Participants consumed significantly more (10 g/67%) from the smaller multipacks than larger pack, which was contributed by those with activated diet consciousness. When diet consciousness concern was activated, participants consumed significantly more (29 g/161%) from the smaller multipacks than larger pack. No effect was found when diet consciousness was not activated. No significant effect of package size was found among those with higher diet consciousness.
Holden, 2015 Study 2 [53] Australia High	114 university students (64 females)	Laboratory face-to-face	M&M’s One 200 g pack Four 50 g packs	Diet consciousness ^5^ Diet prime (food focus) ^6^	Consumption (direct weighing)	Diet consciousness was activated in all participants, no significant effect of package size on consumption was found. Food-focused diet prime was a moderator. Participants’ tendency to overconsume from the smaller multipacks disappeared when food-focused diet prime was provided prior to eating.
John, 2017 Study 1 [38] The USA Low	362 drink purchasers (out of 623 participants) Mean age 24 years	Laboratory computer-based	Sugary drinks (iced tea or lemonade One 680 mL cup Two 340 mL cups One 454 mL cup (control)	None	Consumption (direct weighing) Likelihood of purchase (computer task)	No significant effect of package size on consumption was found. Participants in the two 340 mL cups condition had a significant higher likelihood of purchase compared to those in the one 680 mL cup condition.
Kerameas, 2015 Study 1 [54] Australia Low	87 female undergraduate students Mean age 20 years	Laboratory face-to-face	Cookies 30 g or 90 g total serving size: One 30 g/90 g cookie in one larger bag Three 10 g/30 g cookies in three smaller bags	Perceived norm of appropriate intake ^7^	Consumption (direct weighing)	Participants consumed significantly less from the multiple smaller packages (17 g/24%) than a larger pack. Participants in the 30 g total serving size conditions (additional cookies were available) consumed significantly less compared to those in the 90 g total serving size conditions. The perceived norm of appropriate intake was a mediator. Participants reported a lower perceived norm of appropriate intake when served the multiple smaller packages than a larger package.
Mantzari, 2017 [32] The UK Low	16 household representatives (12 females) Mean age 33 ± 6.6 years	Free living	Cola 1500 mL, 1000 mL, 500 mL, 250 mL bottles	None	Consumption (recording empty bottles)	No powered significance testing was undertaken as it was a feasibility study. The average weekly household consumption when provided with 250 mL, 500 mL, 1000 mL, and 1500 mL bottle size was 7878 ± 3861 mL, 8595 ± 3559 mL, 8331 ± 3963 mL, 8010 ± 3977 mL, respectively.
Mantzari, 2020 [55] The UK Low	16 households Mean age 40 ± 2.7 years	Free living	Wine 750 mL bottles 375 mL bottles	The order of receiving each package condition (crossover)	Consumption (recording empty bottles)	No powered significance testing was undertaken as it was a feasibility study. Household consumption in 2 weeks was 8.4 mL lower when receiving smaller bottles than when receiving larger bottles. The order of receiving each package condition could be a possible moderator. In four weeks, households receiving smaller bottles first overall consumed 1020 mL less wine than those receiving the larger bottles first.
Raynor, 2007 [56] The USA High	24 adults (12 female) Mean age 20 ± 1.6 years	Free living	A snack box with potato chips, crackers, mini cookies, M&M’s 142–227 g packs with smaller/larger total serving size 28–48 g packs with smaller/larger total serving size	Gender Weight	Consumption (recording empty packages)	The total serving size had a significant effect on consumption, regardless of package size. No significant effect of package size on consumption was found. Gender and weight were not moderators.
Roose, 2017 Study 2 [57] Belgium High	188 university students (88 females) Mean age 22 years	Laboratory face-to-face	Brownies One larger bag of 6 brownies Three smaller bags (2 brownies per bag)	Self-control conflict ^8^ Dietary restraint status ^9^	Consumption (direct weighing)	Participants consumed significantly more (13 g/30%) from smaller multipacks than from a larger pack, which was fully contributed by restrained eaters. The self-control conflict was a mediator. Participants experienced less self-control conflict when consuming from the smaller multipacks than a larger pack.
Scott, 2008 Study 2 [18] The USA High	343 university students	Laboratory face-to-face	M&M’s One 200 kcal pack regular-sized M&M’s Four 50 kcal pack of mini M&M’s	Dietary restraint status	Consumption (direct weighing) Perception of energy content (questionnaire)	Participants consumed significantly less from smaller multipacks with mini M&M’s than a larger pack with regular-sized M&M’s. Unrestrained eaters consumed significantly less (48 kcal/38%) from smaller multipacks with mini M&M’s than a larger pack with regular-sized M&M’s. Restrained eaters tended to consume more (12 kcal/12%) from smaller multipacks than a larger pack (not statistically significant). Participants perceived the energy content of smaller multipacks to be significantly greater than that of a larger pack; they also perceived mini M&M’s in smaller multipacks to be more similar to diet foods than regular-sized M&M’s in a larger pack.
Scott, 2008 Study 3 [18] The USA High	96 undergraduate students	Laboratory face-to-face	Cookies One 240 kcal pack regular-sized cookies (4 pieces per pack) Four 60 kcal pack mini cookies (2 pieces per pack)	Dietary restraint status	Consumption (direct weighing) Perception of predicted consumption (questionnaire)	No significant package size effect (mini cookies in smaller multipacks vs. regular-sized cookies in larger pack) was found. Participants predicted that they would consume less from smaller multipacks than a larger pack. Dietary restraint was not a moderator.
Scott, 2008 Study 4 [18] The USA High	393 undergraduate students	Laboratory face-to-face	M&M’s One 200 kcal pack regular-sized M&M’s Four 50 kcal packs mini M&M’s	Dietary restraint status Diet prime (food-focus) ^10^	Consumption (direct weighing) Perception (perceived caloric content) (questionnaire)	Participants consumed significantly less from smaller multipacks with mini M&M’s than a larger pack with regular-sized M&M’s. Participants perceived smaller multipacks with mini M&M’s to be significantly more similar to diet food and had higher energy content than a larger pack with regular M&M’s. Food focus was a moderator for restrained eaters but not for unrestrained eaters. Restrained eaters consumed less from the smaller multipacks than larger pack when regarding the provided snacks as ‘non-food objects’, whereas they consumed more from the smaller multipacks than larger pack when there was no food focus (control).
Stroebele, 2009 [58] The USA High	59 (41 females) Mean age 37.3 ± 12.0 years	Free living	Crackers, chips, biscuits, cookies Four packs (187–360 g per pack) Accordingly, number of smaller packs (19–26 g per pack) to keep the total serving size consistent	The order of receiving each package size condition (crossover)	Consumption (self-recorded snack diary)	On a weekly basis, participants consumed significantly less (187 g/32%) from smaller multipacks than larger packs. Participants who received smaller multipacks first consumed significantly less snacks (28%) from larger packs later, compared to those who received larger packs first and smaller multipacks later.
Van Kleef, 2014 Study 3 [59] The Netherlands High	165 university students (104 females) Mean age 21 ± 2.4 years	Laboratory face-to-face	Mars chocolate bars, package present or absent Three 51 g bars Fifteen 10 g bars	Perception of impulsiveness ^11^ Weight	Consumption (direct weighing) Perception (satiety) (questionnaire)	Participants consumed significantly less (51 kcal/23%) from smaller multipacks than larger packs. The perception of impulsiveness was a mediator. Participants counteracted the feelings of impulsiveness by eating less from smaller multipacks. Larger packs were perceived to be more satiating than smaller multipacks. Weight was not a moderator.
Wansink, 2011 [60] The USA High	37 university students (15 females) Mean age 20.3 ± 1.1 years	Laboratory face-to-face	Crackers One 400 kcal pack Four 100 kcal packs	Weight (mean 23.8 ± 3.9 kg/m^2^)	Consumption (direct weighing)	Participants consumed significantly less (75 kcal/25%) from smaller multipacks than a larger pack, which was fully contributed by overweight participants. No effect was found in normal weight participants. No significant effect of package size on feeling of fullness between package size conditions was found after consumption.
Bui, 2017 Study 1 [49] The USA Low	77 postgraduate students (44 females) Mean age 31 years	Laboratory computer-based	Bite-sized chocolate chip cookies One larger pack (16 pieces per bag) Four smaller packs (4 pieces per bag)	None	Intention to consume (computer task)	No significant effect of package size on intended consumption was found for cookies (which were perceived as an ‘unhealthy food’).
Bui, 2017 Study 2 [49] The USA High	171 (103 females) Mean age 38 years	Laboratory computer-based	Bite-sized chocolate chip cookies Two larger packs (8 pieces per bag) Four smaller packs (4 pieces per bag)	None	Intention to consume (computer task)	No significant effect of package size on intended consumption was found for cookies (which were perceived as an ‘unhealthy food’).
Scott, 2008 Study 3 follow-up [18] The USA High	201 undergraduate students	Laboratory face-to-face	M&M’s One 200 kcal pack regular-sized M&M’s Four 50 kcal packs mini M&M’s	Dietary restraint status	Intention to consume (questionnaire)	Participants intended to eat significantly less (23%) from the smaller multipacks with mini M&M’s than from larger pack with regular-sized M&M’s. Dietary restraint status was not a moderator. However, restrained eaters perceived that considering the consumption of mini M&M’s from smaller multipacks to be significantly more stressful than eating regular-sized M&M’s from a larger pack. This effect was not observed in unrestrained eaters.
Mantzari, 2018 [31] The UK Low	16 household representatives (12 females) Mean age 33 ± 6.6 years	Free living	Cola 1500 mL, 1000 mL, 500 mL, 250 mL bottles	None	Perception of previous consumption (rate and amount) (interview)	Participants believed that their consumption rate and amount was higher with the smallest (250 mL) bottle size due to the perception of more convenient, reduced awareness of the amount consumed, harder for consumption monitoring, and insufficient quantity in each bottle.
Scott, 2008 Study 1 [18] The USA High	385 undergraduate students	Laboratory face-to-face	M&M’s One 200 kcal pack regular-sized M&M’s One 200 kcal pack mini M&M’s Four 50 kcal packs regular-sized M&M’s Four 50 kcal packs mini M&M’s	None	Perception (diet food characteristics and energy content) ^12^ (questionnaire)	Participants perceived that mini M&M’s in smaller multipacks contain significantly more energy (144 kcal/75%) than regular-sized M&M’s in a larger pack. Participants perceived that mini M&M’s in smaller multipacks to be significantly more similar to ‘diet food’ than regular-sized M&M’s in larger packs.
Van Kleef, 2014 Study 2 [59] The Netherlands High	124 university students (75 female)	Laboratory face-to-face	Mars chocolate bars One 51 g pack Five 10 g packs	None	Perception (perceived energy intake) (questionnaire)	Participants overestimated their energy intake more significantly when eating from smaller multipacks (43% more) than a larger pack (4% more). Participants perceived that finishing the provided chocolates in smaller multipacks as significantly less appropriate, more excessive and more impulsive, and resulted in significantly lower expected satiation and satiety than finishing those provided in a larger pack.

^1^ Appearance self-esteem (ASE): the self-worth a person derives from his or her body-image and weight. ^2^ Self-regulatory concern activated group: participants were instructed to complete a body image satisfaction scale and dieting scale and report their weight before the study; self-regulatory concern-inactivated group (control group): participants participated in an unrelated study before the study. ^3^ Diet consciousness-activated group: participants were instructed to complete a body image questionnaire, self-reported height, and weight before the study to manipulate diet consciousness; Diet consciousness-inactivated group (control group): the same questionnaire was given to participants but after food exposure. ^4^ Measured diet-consciousness: participants were categorised into high and low diet consciousness groups by a dietary restraint scale questionnaire. ^5^ Diet consciousness was activated in all participants; participants were instructed to complete a body image questionnaire and report height and weight before the study. ^6^ Food focus: participants were instructed to evaluate the M&Ms while eating. ^7^ Perceived norm of appropriate intake: the perception of appropriate serving size (the appropriate amount of food to consume per eating occasion). ^8^ Self-control conflict: the offer of tempting food to a consumer who is occupied with restraining food intake (i.e., commitment to a health goal) sparks a self-control threat that evokes feelings of conflict. This conflict experience operates as an alarm that signals the need to restrain food intake. Failing to evoke this conflict leads to a failure to exert self-control, which then contributes to overconsumption. ^9^ Dietary restraint status: linked with individual’s perceived ability to estimate energy in this study. Restrained eaters perceived that they have strong ability to determine energy estimation. Unrestrained eaters perceived that they lack ability to determine energy estimation. ^10^ Diet prime conditions: (1) food-focus: participants were instructed to ‘think about the sensory experience of enjoying M&Ms’ such as the texture and taste; (2) non-food focus: participants were instructed to ‘think about the M&Ms as ‘non-food objects’; (3) control condition (no food focus): participants were instructed to ‘think about anything you would like to think’. ^11^ Perception of impulsiveness: participants were instructed to consider the amount of chocolate they consumed and answer five questions on self-perceived impulsiveness (for example, ‘… am self-indulgent’, ‘… cannot resist the temptation of chocolate’). ^12^ Diet food characteristics: measured by 7-point scale on the extent to which they disagree/agree with the statement ‘Overall, the M&Ms in their packages seemed similar to diet foods’.

**Table 4 nutrients-14-00009-t004:** Summary of intervention studies comparing packaged versus unpackaged energy-dense, nutrient-poor snack containing same total serving size.

First Author, Year of Publication, Country, Risk of Bias	Study Sample	Setting	Package Size Comparison Groups	Potential Moderators or Mediators	Outcome Measures (Measures Used)	Findings
Argo, 2012 Study 1 [48] Canada Low	76 female undergraduate students	Laboratory face-to-face	Gumdrops 85 g loosely in a bowl Five 17 g small packs in a bowl	Appearance self-esteem (ASE)	Consumption (direct weighing)	Participants consumed significantly more when snacks were in small packages compared to when snacks were loose, which was fully contributed by those with low ASE (22 g/129% more from packaged than loose snacks). No effect of the presence of small package on consumption was found among participants with high ASE.
Argo, 2012 Study 4 [48] Canada Low	297 female undergraduate students	Laboratory face-to-face	Candy-coated chocolates (88 chocolates) A bowl of loose chocolates Eight small packs	ASE Energy information labelling	Consumption (direct weighing)	Participants consumed significantly more when snacks were in small packages compared to when snacks were loose, which was fully contributed by those with low ASE (28 g/350% more from packaged than loose snacks). No effect of the presence of small package on consumption was found among participants with high ASE. Participants with low ASE consumed significantly more from packaged snacks when they were informed the energy content of small packages was low (compared with when they were informed the energy content was high or when no energy content information).
Argo, 2012 Study 5 [48] Canada Low	105 female undergraduate students	Laboratory face-to-face	Candy-coated chocolates (88 chocolates) A bowl of loose chocolates Eight small packs	ASE Cognitive load (memorising numbers) ^1^	Consumption (direct weighing)	Participants with low ASE consumed significant more (17 g/81%) when snacks were packaged than when snacks were loose. Participants in the low cognitive load condition consumed significantly more (14 g/74%) from snacks that were packaged than from snacks that were loose.
Chance, 2014 Study 2 [61] The USA High	Office kitchen of a technology company	Free living	M&M’s Loose M&M’s in a bulk container M&M’s in small fun packs	None	Consumption (direct observation by trained research assistants)	Participants consumed significantly less (178 kcal/58%) on each occasion when snacks were in smaller packages (fun packs) than when snacks were loosely in the bulk container.
Knowles, 2020 Study 1 [62] The UK Low	80 university students (68 females) Mean age 21 years	Laboratory face-to-face	Brownies Unwrapped in a transparent bowl Wrapped individually in plastic film in a transparent bowl	Perceived effort ^2^ Visual salience ^3^	Consumption (direct weighing)	Participants consumed significantly less when snacks were individually wrapped than when snacks were unwrapped. The perceived effort was a moderator. Unwrapped snacks required less perceived effort to attain than wrapped snacks. The visual salience was a moderator. Unwrapped brownies had a higher visual salience than wrapped brownies.
Cheema, 2008 Study 1 [63] The USA High	22 female undergraduate students	Free living	Chocolates (6 pieces in a box) Unwrapped Wrapped individually in foil	Self-regulatory concern (aversion to overconsume)	Rate of consumption (self-reported response sheet)	All participants were required to finish provided chocolates in a week. Participants consumed wrapped chocolates significantly more slowly than those that were unwrapped (consumed 45 out of 66 pieces in total if wrapped vs. 60 out of 66 in total if unwrapped, in first two days). This effect was fully contributed by participants who had greater self-regulatory concerns. No significant effect was found in participants with no self-regulatory concern.
Cheema, 2008 Study 4 [63] The USA High	54 university students	Free living	Cookies (20 pieces per condition) Unwrapped Wrapped individually in white wax paper Wrapped individually in different colour	Package colour	Rate of consumption (direct observation)	Participants consumed cookies that were individually wrapped in coloured packages significantly more slowly than those that were individually wrapped in white packages, or those that were unwrapped. Participants with cookies wrapped in white packages had the same consumption rate as participants with unwrapped cookies. A total of 17 of 20 participants finished cookies that were individually wrapped in coloured packages, all 20 participants finished cookies were individually wrapped in white packages or unwrapped.

^1^ Cognitive load: participants were given a memory task (memorising numbers) at the start. Participants in low cognitive load condition were required to memorise a two-digit number, whereas those in high cognitive load condition were required to remember an eight-digit number. ^2^ Perceived effort: the required effort to attain the provided snack. ^3^ Visual salience: the subjective perception of attractive properties of the provided snack.

#### 3.3.1. Smaller versus Larger Single Package with Different Total Serving Size

Out of 47 intervention studies, 15 quantitatively examined the effects of a smaller versus larger single pack of snacks or drinks (Table 2). The majority of outcome measures was consumption (*n* = 9), followed by intention to consume (*n* = 6), intention or likelihood of purchase (*n* = 3), and perception (*n* = 2). Out of these 15 studies, 12 found a significant relationship between package downsizing and a lower actual or intended consumption. Two studies in children aged between 3 and 7 years [37] and six studies in adults reported a significantly lower consumption when snacks or drinks (including alcoholic and non-alcoholic drinks) were provided in a smaller compared to a larger package [32,38,39,40,42,43]. Four studies found adult participants’ intention to consume was significantly lower when snacks were provided in a smaller than a larger package [41,45,46]. Participants’ intention or likelihood of purchase was not affected by snack or drink package size in all three studies [38,47]. Two studies assessing perception found that participants considered popcorn served in a smaller package to be ‘healthier’ than when served in a larger size [42], and they believed the unit price to be higher when the package size became smaller [46].

The studies in Table 2 differed considerably in their design. A few studies selected much larger EDNP serving sizes than reasonable for a single eating occasion. Five studies using M&M’s chocolates provided a small package size containing more than 100 g and a large package size greater than 300 g [39,41,45,46]. Amongst these, four studies observed a significant package size effect [39,41,45,46] whilst one found no effect [40].

Twelve out of 15 studies examined the effect of potential moderators, including gender (*n* = 8), food preference (*n* = 5), prior exposure to diet-related materials (*n* = 2), dietary restraint (*n* = 2), and serving size labelling (*n* = 2). Three studies found that the effects of a smaller single package on reducing consumption or intention to consume was more prominent among males than females [40,45], but five other studies did not observe any difference between genders [37,41,44,47]. Four studies using popcorn observed the tendency to overeat from a larger package was more prominent when children [37] and adults [42,43] were offered their preferred snacks (for example, fresh popcorn) compared to not preferred (for example, stale popcorn); however, this was not observed in one study in adults [39]. Compared with exposure to non-diet-related material (such as a travel magazine), exposure to diet-related material (such as a weight loss and fitness magazine) prior to eating resulted in reduced intake from the larger single package and diminished the package size effect [41]. Two studies reported dietary restraint was not a moderator of the package size effect [40,41]. The pictorial serving size recommendation labelling moderated the package size effect by lowering participants’ intention to overconsume from larger, but not from smaller packages in one study [45], but this was not observed in another study [45].

#### 3.3.2. Smaller Multipacks versus Larger Package(s) with Same Total Serving Size

Out of the 47 intervention studies, 25 involved interventions that used smaller multipacks versus a larger package (or larger packages) with the same total serving size (Table 3). Of these, 24 were quantitative studies. Outcome measures were mostly consumption (*n* = 19), followed by perception (*n* = 7), intention to consume (*n* = 3), and likelihood of purchase (*n* = 1). These studies varied considerably in total serving size, ranging from 30 g to 400 g. The relative size difference between the smaller multipacks and larger packages was also large, and there was a greater than four-fold difference in many studies [52,56,58,59].

Eleven out of 19 studies assessed the presence of the package size effect on consumption among all participants. Amongst these, six studies found that snacks served in smaller multipacks resulted in significantly lower consumption compared with a larger pack [18,50,58,59,60], while five studies did not observe any significant effect [18,38,49,51,56]. Studies examining the intention to consume reported mixed results, with one study showing that smaller multipacks led to lower intention to consume [18], whereas two other studies did not observe any difference in intention to consume between larger pack(s) and smaller multipacks [49]. One study noted that smaller multipacks had a significantly higher likelihood of being purchased compared to a larger pack, although the actual consumption did not differ between package size conditions [38].

Seven studies investigated participants’ perceptions regarding consuming EDNP snacks and drinks from smaller multipacks compared to larger packs [18,31,59]. The qualitative study noted participants believed that the amount and rate of consumption of sugar-sweetened drinks to be higher when receiving smaller (250 mL) compared with larger bottles (1500 mL) over a two-week period [30]. Participants also tended to believe that smaller multipacks contained significantly more energy and eating from smaller multipacks was ‘less appropriate’ and ‘more excessive’ than from larger pack(s), even though the total serving size was kept constant between package size conditions [18,59]. Participants predicted that they would eat less from smaller multipacks and overestimated their energy intake to a higher extent when eating from smaller multipacks compared with the larger pack [59]. However, they also perceived that snacks provided in smaller multipacks would result in lower satiety and were more similar to ‘diet food’ than those in a larger pack [18].

Eighteen out of 25 studies reported potential moderators of the package size effect, including dietary restraint status (*n* = 6), body weight (*n* = 4), gender (*n* = 3), order of exposure (*n* = 3), diet consciousness (*n* = 2), food-focused diet prime (*n* = 2), appearance self-esteem (ASE) (*n* = 1), self-regulatory concern (*n* = 1), and package design (*n* = 1); mediators, including the perception of impulsiveness (*n* = 1), perceived norm of an appropriate intake (*n* = 1), and self-control conflict (*n* = 1). Three out of six studies noted smaller multipacks led to increased consumption compared with the larger pack among those with high dietary restraint status, while this was not observed in unrestrained eaters [18,57]. Two out of four studies found overweight and obese participants consumed significantly less from smaller multipacks than the larger pack, while no effect was observed in healthy-weight participants [52,60]. Gender was not shown to be a moderator in the three studies that specifically assessed gender [48,49,56]. Regarding the order of exposure, one crossover study reported that participants who received smaller multipacks in the first week consumed significantly less from larger packs in the second week compared to those who received the larger pack first [58]. However, this moderating effect was not observed in the other two studies [50,55]. Two studies found those with inactivated or low diet consciousness had a significantly higher tendency to overconsume from smaller multipacks than larger packs [53]. A food-focused diet prime acted as a moderator in two studies. In one of these, the exposure to a food-focused diet prime prior to eating (that is, participants were instructed to evaluate the provided food) eliminated the tendency to overconsume from smaller multipacks [53]. However, in the other study, contradictory results were shown for restrained eaters, who consumed less from smaller multipacks than from the larger pack only when exposed to a non-food-focused diet prime (being instructed to regard the provided snacks as ‘non-food items’) [18]. Smaller multipacks may lead to increased consumption compared with larger packs among participants with low ASE, but did not have any effect on those who had high ASE [48]. The activation of the self-regulatory concern (by completing a body satisfactory scale and reporting weight before the study) moderated the package size effect by reducing consumption from larger packs, but not from smaller multipacks [51]. When the package design was transparent (compared to opaque packages), participants ate more from smaller multipacks than larger packs [48]. The examination of mediators showed that the perception of impulsiveness [59], perceived norm of an appropriate intake [54], and self-control conflict [57] were all significant mediators in the relationship between package size and outcome measures. Smaller multipacks contributed to reduced consumption as participants experienced lower perceived norm of appropriate intake and higher perceived impulsiveness when eating from smaller multipacks compared to larger pack(s) [54,59]. Conversely, one study that found smaller multipacks was associated with increased consumption, reported that participants experienced lower self-control conflict when consuming from smaller packages rather than larger ones [57].

#### 3.3.3. Packaged versus Unpackaged Snacks with the Same Total Serving Size

In total, seven out of the 47 intervention studies compared the effect of packaged and unpackaged (unwrapped or loose) EDNP snacks with the same total serving size. The outcome measures included consumption (*n* = 5) and the rate of consumption (*n* = 2). Two studies found that participants consumed significantly less when snacks were wrapped in small packages compared with loose snacks [61,62]. Examining the rate of consumption, one study informing participants to finish all provided snacks, found that snacks wrapped in individual smaller packages resulted in a significantly slower rate of consumption than snacks that were unwrapped [63]. Another study noted that this rate-lowering effect was only significant when snacks were in coloured small packages, with less participants finishing snacks in coloured small packages compared to those in white-coloured small packages or unpackaged [63].

Six out of seven studies found that the association between the presence of packaging and outcome measure was influenced by moderators, including ASE (*n* = 3), cognitive load (*n* = 1), energy information labelling (*n* = 1), package colour (*n* = 1), perceived effort (*n* = 1), self-regulatory concern (*n* = 1), and visual salience (*n* = 1). In three studies, participants with a low ASE consumed significantly more from snacks that were wrapped in small packages than those that were unpackaged [48], especially when the energy information labelling on packaging indicated a low energy content [48]. One study observed that participants who had low cognitive loads (that is, when their cognitive resources were not occupied by another task) consumed significantly more snacks when they were individually wrapped than those that were unpackaged [48]. Another study noted participants with higher self-regulatory concern spent more time finishing chocolates that were individually wrapped than those that were loose [63]. In contrast, no significant effect of the presence of packaging (compared to unwrapped snacks) on consumption or the rate of consumption was found among those who had high ASE, high cognitive loads, or no self-regulatory concern [48,63]. Compared with unwrapped snacks, attaining snacks that were wrapped in individual small packages was associated with higher perceived effort, leading to lower consumption [62]. Snacks that were individually wrapped in coloured packages led to a slower consumption rate compared to unpackaged equivalents; however, this effect did not persist when snacks were wrapped in white-coloured packages [63].

## 4. Discussion

This scoping review aimed to assess the effects of reducing the package size of EDNP snacks and drinks on consumption, intentions, and perception, and examine the effects of potential moderators or mediators. Three types of interventions were identified: a single smaller versus single larger package (different serving size), multipacks versus larger packages (same total serving size), and small packaged versus unpackaged (same total serving size). Overall, package size seemed to be a strong environmental determinant of EDNP snack and drink consumption, intention to consume, and/or consumer’s food-related perception. Most studies observed that reducing the package size and total serving size simultaneously (that is, single smaller versus single larger package) led to significantly lower actual or intended consumption of EDNP snacks and drinks. Findings were inconsistent in studies that reduced package sizes while keeping the total serving size constant between package size conditions (that is, smaller multipacks versus larger packages or unpackaged snacks). The presence of multiple moderators, in particular for the studies using multipacks, added complexity to the interpretation of the results.

The package size effect was robust when comparing a single smaller versus a single larger package, supporting the well-known ‘portion size effect’ that people tend to unconsciously eat more when exposed to larger serving sizes [19]. This finding was consistent with the Cochrane review from Hollands and colleagues that larger package sizes result in higher intakes in both adults and children [14]. A number of mechanisms have been proposed to explain this effect, the ‘appropriateness mechanism’ and ‘unit bias’ are the most prevalent explanations [13,64,65]. When the serving size is within reason, people tend to rely more on external cues such as serving size rather than internal cues such as satiety to determine the amount of intake; they are also likely to regard a single package (that is, a single unit) to be an appropriate amount to consume [13,15,64]. Frequent exposure to a single smaller package could shift the norm, or the appropriate amount of intake, downwards and, thus, facilitate a better portion control [64,65]. We found that participants still tended to consume more from a larger package even when the single package was oversized (for example, more than 100 g chocolate in a single smaller package and more than 300 g in a single larger package) [39,41,45,46]. This indicated that the portion size effect persisted even when the serving sizes presented were much larger than reasonable. We acknowledge that it was conceptually impossible for participants to consume more from a single smaller package than a larger package as not all laboratory studies offered additional food in the smaller package size condition. However, when considering a real-life environment (for example, at a cinema), selecting or extracting an appropriate portion size from a large package can be challenging [13].

The evidence for a ‘multipack effect’, that is, when the total serving size was kept consistent, the effect of smaller multipacks compared to larger packages or unpackaged loose snacks on consumption, intention to consume, or purchase was equivocal. Less than half the studies examined the presence of the package size effect in all participants without considering moderators or mediators; the majority of these found a significant effect of smaller multipacks in reducing consumption or intention to consume compared to larger packages or unpackaged foods [18,50,58,59,60]. The underlying mechanism of this multipack effect when dividing a fixed amount of food (in larger packages or unpackaged) into multiple smaller packages has not been clearly documented. A possible explanation might be the ‘segmentation effect’, whereby consumption of multiple smaller packages, rather than one larger package, is perceived to be less appropriate, even though the serving size is fixed [54,64,66]. The feeling of ‘inappropriateness’ may be due to the process of unwrapping multiple smaller packages that can interrupt the mindless automated eating episodes by providing opportunities for a pause [66]. Nevertheless, a few studies did not observe a significant multipack effect [18,38,49,51,56]. The numerous variations in study design, including the selected food type, package size, total serving size, and the presentation of food packages (for example, number of packages, packaging shape, colour, and on-pack illustrations), as well as different moderators, can all influence study outcomes [13,14,67].

Most of the included studies assessed the effects of internal moderators or mediators (that is, individual characteristics or psychological factors that cannot be easily changed), whereas the effects of external moderators (that is, factors that can be manipulated by environmental intervention) have not been well studied. Gender, dietary restraint status, food preference (or food liking), body weight, and order of exposure were the most frequently examined moderators, but many others were assessed in only one or two intervention studies. The moderating effects of body weight, gender, and order of exposure were not consistently significant across studies. Food preference appeared to moderate the package size effect in both children and adults. Exposure to preferred snacks resulted in an increased tendency to overeat from larger packages compared to smaller packages. In addition, most studies that assessed internal factors, including dietary restraint status and ASE, reported a significant moderating effect. Those with higher dietary restraint or lower ASE tended to eat significantly more from smaller packages than larger packages. Previous studies have shown that restrained eaters are more responsive to food cues than unrestrained eaters [68] and, therefore, restrained eaters may respond differently to package downsizing than unrestrained eaters. More evidence is needed to elucidate the role of internal and external factors on the package size effect, and to prevent any unintended compensatory eating behaviours. In particular, the impact of external moderators, such as the country of residence, accessibility, cost, and package design, remains unclear.

### Strength and Limitations

Strengths of the review include searching of multiple databases of both published and grey literature. Double screening and data extraction were performed to ensure consistency, and the quality of each eligible study was assessed using a reputable critical appraisal tool. We also acknowledge several limitations in this review with study populations being highly homogeneous, and predominantly convenient samples of university students or young adults who were more likely to attain higher levels of education. All studies were conducted in high-income western countries, in the US, Canada, Europe, and Australia. No cross-country differences were detected, but the majority of studies was conducted in the US, and only a small number in some European countries. In addition, the number of studies, in particular of multipacks and unpackaged versus packaged, was relatively small, although an increase over time was noted, especially after the Cochrane review in 2015, clearly demonstrating the portion size effect [14] and the WHO recommendations (2014) to limit food portion size as a strategy to reduce energy intake [22,69].

Included studies mostly investigated a single eating occasion in a laboratory setting. No medium- or long-term data for package downsizing were available, and no study assessed compensatory eating behaviours at subsequent eating occasions. Although many internal moderators were identified, most were only assessed in one or two intervention studies. Very few external moderators were examined.

Our assessment of the quality of the included studies revealed a mix of a low and high risk of bias studies for each of the three types of package size interventions. Out of 47 studies, all except for three low-risk and two high- risk studies reported significant findings. Most studies had reliable outcome measures and appropriate statistical analysis, but not all RCTs provided sufficient information on the randomisation and blinding method, and most quasi-experimental studies did not have a control group and/or failed to identify potential confounders (for example, body weight) between comparison groups.

## 5. Conclusions

This scoping review assessed a range of package sizing interventions using EDNP snacks and drinks; examining smaller packages versus larger packages containing different amounts, as well as smaller multipacks versus larger packages (or those unpackaged) that contained the same amount. We found that rather than providing multipacks, an overall reduction of a single package size was a more promising strategy to reduce the consumption of EDNP snacks and drinks. Exposure to smaller amounts of food resulted in consumers eating less and potentially preventing the likelihood of mindless eating. Therefore, the availability of smaller single packages acceptable to consumers can encourage better selection of appropriate portion sizes, and continuous exposure over time will recalibrate the portion size norm towards smaller sizes [70]. Our understanding of the multipack effect is limited by the presence of multiple moderators and mediators. Some studies have shown multipacks may facilitate overconsumption among certain subgroups, such as those with higher dietary restraint or lower ASE. It is not known whether compensatory behaviour occurs at subsequent meals in response to extreme package size reduction. Longer-term high-quality studies with a more representative study sample are needed to ensure there are no unintended consequences.

We acknowledge that package downsizing is only one of many public health strategies aimed at reducing EDNP snack and drink intake and slowing the progression of excessive weight gain, and other strategies such as education, price, reformulation, food labelling, and front-of-pack visual cues are also necessary [18,66,69]. This review provides evidence supporting recommendations to change the current food environment to promote single smaller packages and restrict the accessibility to larger packages of EDNP snacks and drinks [19,71,72]. Active engagement of the food industry, as well as coordination between stakeholders (for example, policy makers, food manufacturers, retailers, and consumers), are crucial for modifying the food environment to encourage more appropriate portion size selections [19,73]. However, the acceptability and feasibility of package size reductions by consumers and the industry is unknown, as is the optimal serving size for the various EDNP snacks and drinks. A recent analysis found that the food industry has responded to consumer demand by introducing smaller package sizes of carbonates and confectionery foods, as reflected in their sale trends [74]. Further research is required to improve our understanding of appropriate serving sizes of EDNP snacks and drinks, and how reductions in serving sizes can be implemented at a population level [13,19].

## Figures and Tables

**Figure 1 nutrients-14-00009-f001:**
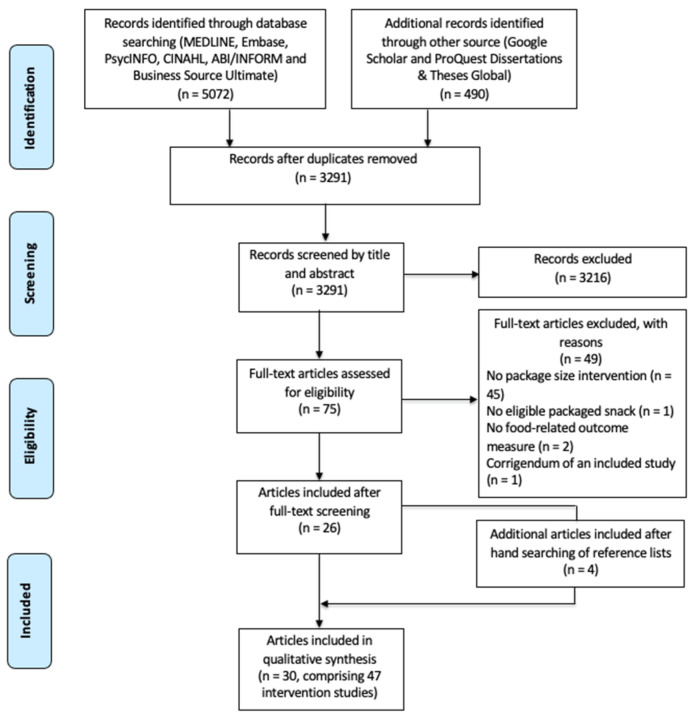
Preferred reporting items for systematic reviews and meta-analyses for the scoping review process (PRISMA-ScR) flow diagram.

**Figure 2 nutrients-14-00009-f002:**
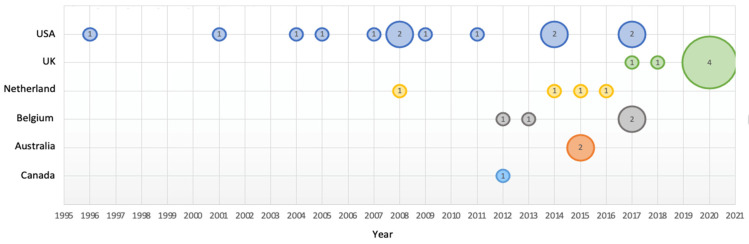
Publication years and countries of included articles (*n* = 30). Countries are represented by different colour. The bubble size is proportional to the number of articles published in the year and the country.

**Table 1 nutrients-14-00009-t001:** Inclusion and exclusion criteria for scoping review.

	Inclusion	Exclusion
Participants	Human participants	N/A
Concept	Energy-dense, nutrient-poor packaged snacks and drinks Exposure to package size reduction or intervention Qualitatively or quantitatively measured consumption, intention to consume or purchase, or perception (for example, the perceived healthfulness of snacks in a smaller package size)	Main meals, fast foods, food from core food groups No exposure to direct package size intervention No measurement of consumption, intention to consume or purchase, or perceptions related to food-choice making
Context	High-to-middle income country All study contexts Studies from selected databases, grey literature	Low-income country Review studies Studies not written in English

## Data Availability

The data presented in this study are available in Appendix A.

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
