# Peer review of "The Effect of Downsizing Packages of Energy-Dense, Nutrient-Poor Snacks and Drinks on Consumption, Intentions, and Perceptions—A Scoping Review"

_nutrients, 2021, doi:10.3390/nu14010009_

Round 1
Reviewer 1 Report
General comments
The topic is relevant in the field of public health strategies. The objective of this scoping review was to systematically map the existing research studies to: 1) assess the effect of reducing the package size of EDNP snacks and drinks (alcoholic and non-alcoholic) on consumption, intention to consume and purchase, and perceptions; and 2) assess the effects of potential moderators or mediators.
There are a few minor points that could be considered:
Introduction
- The Introduction section embraces subjects that are associated with the topic, and is built with actual and appropriate references. Nevertheless, we invited authors to consider the movements that promotes the “eat more” food environment, namely the food industry’s economic imperative to increase sales in a hugely competitive marketplace (e.g.: Neste, M., 2013).
Methods
We are facing a robust research applying a scoping review methodology. Nevertheless, we ask authors to clarify the following points:
- Authors should define in terms of dimensions what they considered as smaller and larger package sizes.
- Can authors clarify the differences between “free-living setting” and “free-naturalistic setting”?
- On p. 6 can authors please identify the number of studies that examined EDNP snacks and drinks: confectionery (n=), biscuits and sweet pastries (n=), popcorn (n=), savoury snacks (n=), sugar-sweetened drinks and alcohol (n=), and other product categories (n=).
Results
- Can authors plot a graph with the distribution of publications per year in the analysis?
- Please include the “setting” variable (face-to-face, computer-based laboratory, free-living, naturalistic settings) in Tables 2-4.
- In the Tables 2-4 the aim of each paper should be included.
- In the Tables 2-4 the way that outcomes were measured should be included. For instance, how “consumption”, “intention to consumption” and other outcomes were measured for each research?
Discussion
Can authors discuss the evolution of the publications by year considered in your sample and explain the reasons for the potential trends?
Neste, M. (2013), Food politics, University of California Press.
Author Response
Please see the attachment, thank you.

Reviewer 2 Report
The topic is of interest for Nutrients’ readers as well as the manuscript is properly written, easy to read, and well organized. The authors, by reviewing studies in the literature, explore the effect of reducing the package size of energy-dense, nutrient-poor (EDNP) snacks and drinks on consumption, intentions, and perception, and to examine the effects of potential moderators or mediators. The methodology used to select the studies fits the scientific standards.
However, besides such manuscript's strengthens, there are also weaknesses that can be fixed in the second round of revisions.
Authors may provide additional discussion on cross-countries differences in findings selected as well as discuss findings according to socio-economic and demographic characteristics of individuals recruited in each study selected. Lastly, I suggest the authors to expand the policy implications of this review.
Author Response
Please see the attachment, thank you.
